



# Multi-decadal Streamflow Projections for Catchments in Brazil based on CMIP6 Multi-model Simulations and Neural Network Embeddings for Linear Regression Models

Michael Scheuerer[1], Emilie Byermoen[2,3], Julia Ribeiro de Oliveira[4], Thea Roksvåg[1], and Dagrun Vikhamar Schuler[3]

[1]Statistical Analysis and Machine Learning Department, Norsk Regnesentral STI, Oslo, Norway
[2]Geophysical Institute, University of Bergen, Bergen, Norway
[3]Water, Weather and Climate Department (MEW), Statkraft Energi AS, Oslo, Norway
[4]Markets Brazil, Statkraft Energia do Brasil Ltda, Rio de Janeiro, Brazil

**Correspondence:** Michael Scheuerer (scheuerer@nr.no)

**Abstract.** A linear regression model is developed to link anomalies of streamflow to anomalies of precipitation amounts and temperature with the goal of making multi-decadal streamflow projections based on CMIP6 multi-model simulations. Regression coefficients estimated separately for each catchment and each month show physically implausible spatial patterns and indicate issues with overfitting. An alternative approach is therefore explored in which all regression coefficients are estimated

simultaneously through a neural network that retains the original linear model structure, but uses embeddings to map each combination of catchment and month to a set of regression coefficients. The model is demonstrated over a set of catchments in Brazil, where the estimated relationships are used to make streamflow projections for the next decades based on CMIP6 multi-model simulations. It yields physically more plausible relationships between streamflow, precipitation amounts, and temperature for our study area than the locally fitted regression models. The resulting projections indicate reduced streamflow

over northern, north-eastern, central, and south-eastern Brazil, especially for the austral spring and summer season. The signal is less clear during austral winter. In southern Brazil, an increase in streamflow is expected.

## 1 Introduction

Brazil is considered to be an important growth region for both wind- and hydropower production and has generated 63%

(over 427.000 GWh) of its electricity in 2022 through hydropower (International Energy Agency, 2022). Statkraft is one of the renewable energy producers who own and operate several hydropower plants in Brazil, and is therefore highly interested in estimates of future streamflow trends in the country. Many catchments in Brazil have experienced a decline in precipitation and streamflow in the past (e.g., Luiz Silva et al., 2019), and hydroclimatological projections point towards reduced and more variable rainfall in the future (Zaninelli et al., 2019; Reboita et al., 2022; Alves et al., 2020). Other catchments, primarily in



southern Brazil, have seen an increase in precipitation and streamflow (e.g., Luiz Silva et al., 2019, their Table 3). These trends
are linked to a southward shifting of the average location of the South Atlantic Convergence Zone (Zilli et al., 2019) and direct
(through reduced runoff) or indirect (through increased atmospheric moisture content) implications of increased evapotranspi-
ration. It is unclear, however, to what degree the observed changes are part of longer, on-going trends or part of multi-decadal
oscillations in the climate system. By analyzing multi-decadal simulations of a wide variety of climate models, e.g. from the

Coupled Model Intercomparison Project Phase 6 (CMIP6, Eyring et al., 2016), one can attempt to obtain projections of the
future potential for hydropower production in Brazil and help authorities and energy companies foresee areas in risk of future
long-term energy shortages. In this study, our aim is to build a relationship between streamflow, precipitation and temperature
for Brazilian catchments and use it to project future monthly streamflow with CMIP6 multi-model simulations as input.

One possible approach to achieve that is to use a process-based hydrological model (Fatichi et al., 2016; Clark et al., 2017).

In Norway, the main focus of Statkraft's operations, the HBV model (Bergström, 1992) and other hydrological models that are
specialized in simulating snow storage and snow melt (e.g., Xu, 2002) are commonly used for simulating streamflow. In Brazil,
evapotranspiration constitutes a significant portion of the water balance, and hydrological models that put more emphasis on
this aspect are needed. Statkraft's in-house hydrological model has not yet been adapted to tropical climates and is therefore
not readily available for use with meteorological forcings derived from climate model output over South America.

An alternative way to simulate streamflow is the use of data-driven methods. Lately, Long Short-Term Memory (LSTM)
networks have achieved notable advancements in the field of rainfall-runoff modeling (Kratzert et al., 2018; Frame et al., 2022;
Arsenault et al., 2023). LSTMs demonstrate strong performance when trained on daily streamflow data, but may also perform
well with monthly data, provided that the monthly records are sufficiently long (Clark et al., 2024). Challenges with LSTM
networks and other AI models arise in connection with explainability, i.e. the ability to understand and trust their decisions

(De la Fuente et al., 2024) and overfitting, i.e. the risk of picking up specific details in the training data which do not generalize
to new, unseen data. The latter can be particularly problematic in the context of modeling rainfall-runoff relationships which
are then applied to climate model output, since the climate projections cover scenarios outside the range of the historically
observed climate. Few studies exist in the literature which use LSTMs for decadal predictions (Slater et al., 2023). In addition,
impacts of deforestation and land use on the hydrological cycle can be quite significant in Brazil (e.g., Baudena et al., 2021;

Caballero et al., 2022; Chagas et al., 2022) and relevant data about these effects are not available. A 'black box' model that may
pick up these changes but lacks information to extrapolate them into the future poses a risk when it comes to generalization to
unseen data.

Considering the above factors and some exploratory data analysis, we decided to use a low-dimensional linear regression
model that builds a statistical relationship between precipitation, temperature and observed streamflow anomalies on a monthly

time scale for each catchment. Unlike more complex machine learning models, this type of model is fully interpretable (e.g.,
Flora et al., 2024, their Fig. 1) and permits an intuitive understanding of how changes in precipitation and temperature affect
streamflow. Fitting a separate linear regression model for each catchment and month, however, resulted in regression coeffi-
cients that were spatially inconsistent over our study area, and constraints had to be imposed to prevent physically implausible
rainfall-temperature-runoff relationships. A variant of the baseline approach is therefore proposed, which employs a neural





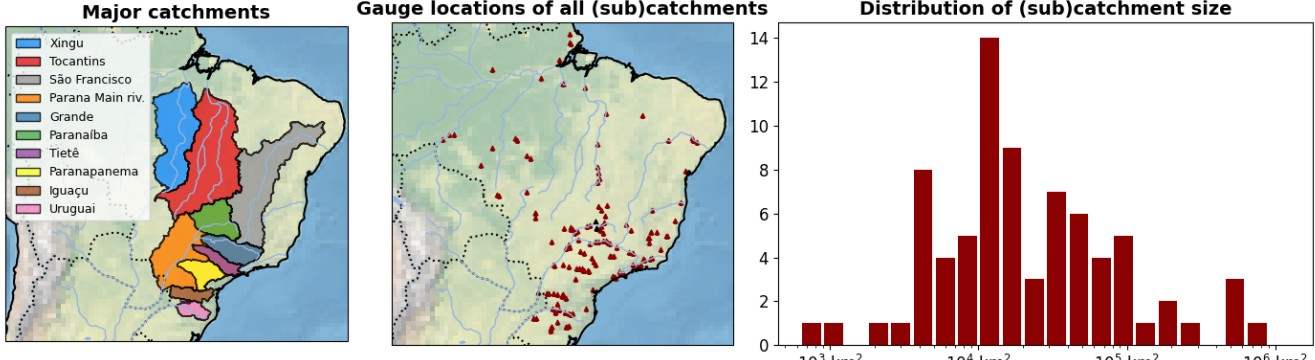

**Figure 1.** Overview over the (sub)catchments and gauge locations considered in this study.

network framework that retains the linear model structure, but uses embeddings (Guo and Berkhahn, 2016) to map each combination of catchment and month to a set of regression coefficients. This permits sharing of information across space and time and in our example yields coefficient patterns that are physically more plausible, even without the use of constraints. The simple structure of the model, makes it well-suited for situations where the data availability is limited, such as when only short records of monthly streamflow data are available and LSTMs may not perform as effectively. The model can also easily be transferred to new regions of the world without additional modeling effort and fine-tuning. To our knowledge, this is the first use of a neural network framework in this particular way, i.e., where its flexibility is exploited to learn complex and non-linear spatio-temporal coefficient patterns while retaining a fully interpretable, linear structure with regard to the primary predictors.

The rest of the paper is structured as follows: Section 2 gives an overview over the data used in this study and presents some exploratory data analysis used to inform subsequent methodological choices. The statistical model itself is introduced in Section 3, first in its basic form as a linear regression model and then in the variant that uses a neural network to represent spatial and temporal patterns of the regression coefficients. Results are presented and discussed in Section 4 and include metrics that assess the quality and limitations of the statistical model as well as streamflow projections obtained with it. Section 5 discusses the issue of uncertainty our projections while Section 6 concludes with a summary and a discussion of the use of the presented methodology.

## 2 Data and Exploratory Analysis

### 2.1 Streamflow data

We use time series of natural total monthly streamflow downloaded through the API of Brazil's National Operator of the Electric System (ONS). To eliminate the challenges posed by non-stationarities in observed streamflow series due to evolving consumptive uses, ONS derives natural streamflows from observed series at river gauging stations by incorporating inflow and discharge at utilization sites while accounting for reservoir operations upstream, consumptive uses, and net evaporation



(Operador Nacional do Sistema Elétrico, 2018). For this project, a subset of 157 Brazilian gauge locations was used for which we have mostly complete monthly streamflow series during the period from 1960 to 2020.

Figure 1 gives an overview over the gauge locations and associated catchments considered here, and shows that catchments from all regions within Brazil are represented with areas varying from a few hundred square kilometers to several hundred of thousands of square kilometers. Many of these catchments are nested, i.e. streamflow is measured at several points of a river and its tributaries, and in each case the associated catchment is taken to be the area over which water flowing through this point is collected.

## 2.2 Precipitation and temperature data

As a proxy for local rainfall amounts we use the Climate Hazards group InfraRed Precipitation with a Station dataset (CHIRPS) version 2.0 (Funk et al., 2015), which was downloaded at monthly temporal resolution and $0.05°$ horizontal resolution and up-scaled to $0.25°$ resolution before further processing. This data product is constructed by combining in-situ station observations with satellite precipitation estimates in order to represent sparsely gauged regions. It was found to agree well with observations across all regions in Brazil with some lower similarity over the Northwest of Amazon and the southwest of Pará state (Costa et al., 2019). As a consequence of being based on satellite data though, the CHIRPS product is only available from 1981 onwards, which makes it the limiting factor in our setup regarding training sample size.

The average 2-m temperature over each catchment was calculated from the ERA5 dataset (Hersbach et al., 2023), a state-of-the-art reanalysis product made by the European Centre for Medium-Range Weather Forecasts (ECMWF). These data were downloaded from the Copernicus Climate Change Service (C3S) Climate Data Store (CDS) for the 1981-2020 period. Total precipitation accumulation is also available as a variable in ERA5, but station observations of precipitation are not included in the ERA5 data assimilation scheme, and additional analysis (not shown here) suggested that CHIRPS provides a more accurate representation of monthly rainfall over Brazil and was therefore preferred for this variable.

Both CHIRPS precipitation data and ERA5 temperature data were aggregated to the catchment scale by averaging the values across all grid points within the boundaries of each catchments. For very small catchments, the nearest grid point to the catchment area was used.

## 2.3 Climate model data

Simulations of 2-m temperature and precipitation from the Coupled Model Intercomparison Project Phase 6 (CMIP6, Eyring et al., 2016) multi-model ensemble were downloaded from the Earth System Grid Federation (ESGF). The SSP2-4.5 scenario was selected, which assumes a moderate level of greenhouse gas emissions in the calculations of the future precipitation and temperature (O'Neill et al., 2016). The datasets are available for both a historical period (1850-2014) and a projection period (2015-2100). Climate model projections in CMIP6 are aimed at simulating the long-term future climate based on changed boundary conditions and the principles of global energy balance. Due to internal climate variability, even different simulations from a single climate model can yield very different precipitation and temperature profiles in individual years and even decades. To sample this internal climate variability as well as possible, we use output from all available CMIP6 models





**Table 1.** CMIP6 climate models used in this study

| | |
|---|---|
| ACCESS-CM2 (Australia) | Bi et al. (2020); Dix et al. (2019) |
| BCC-CSM2-MR (China) | Wu et al. (2019); Xin et al. (2019) |
| CESM2 (USA) | Danabasoglu et al. (2020); Danabasoglu (2019) |
| CMCC-ESM2 (Italy) | Cherchi et al. (2019); Lovato et al. (2021) |
| CNRM-CM6-1 (France) | Voldoire et al. (2019); Voldoire (2019a) |
| CNRM-ESM2-1 (France) | Séférian et al. (2019); Voldoire (2019b) |
| EC-Earth3-CC (Europe) | Döscher et al. (2022); EC-Earth (2021) |
| GFDL-ESM4 (USA) | Dunne et al. (2020); John et al. (2018) |
| HadGEM3-GC31-LL (UK) | Kuhlbrodt et al. (2018); Good (2019) |
| IITM-ESM (India) | Swapna et al. (2018); Singh et al. (2020) |
| INM-CM4-8 (Russia) | Volodin et al. (2018, 2019a) |
| INM-CM5-0 (Russia) | Volodin et al. (2017, 2019b) |
| IPSL-CM6A-LR (France) | Boucher et al. (2020, 2019) |
| KACE-1-0-G (South Korea) | Lee et al. (2020); Byun et al. (2019) |
| KIOST-ESM (South Korea) | Pak et al. (2021); Kim et al. (2019) |
| MIROC-ES2L (Japan) | Hajima et al. (2020); Tachiiri et al. (2019) |
| MIROC6 (Japan) | Tatebe et al. (2019); Shiogama et al. (2019) |
| MPI-ESM1-2-LR (Germany) | Mauritsen et al. (2019); Wieners et al. (2019) |
| MRI-ESM2-0 (Japan) | Yukimoto et al. (2019a, b) |
| NESM3 (China) | Cao et al. (2018); Cao (2019) |
| NorESM2-MM (Norway) | Seland et al. (2020); Bentsen et al. (2019) |
| UKESM1-0-LL | Sellar et al. (2019); Good et al. (2019) |

which had simulations of both temperature and precipitation over the time period considered here and were used in this study.
110 The resulting selection of 22 models is listed in Table 1. The simulations were aggregated to the catchment scale in the same
way as described above for the CHIRPS and ERA5 data. For reasons further explained in Section 3.1, no downscaling or bias
correction was performed at this stage.

## 2.4 Precipitation and streamflow climatology in different parts of Brazil

The precipitation regime in northern and northeastern Brazil is dominated by the Intertropical Convergence Zone (ITCZ),
115 a belt near the equator associated with heavy precipitation oscillating north- and southwards depending on the position of
maximum incoming solar radiation (Garreaud et al., 2009). In the central part of the country, where several of the large river
systems carrying water northward and southward to hydro-electrical plants are formed, the South Atlantic Convergence Zone
(SACZ) regime dominates (Rosa et al., 2020). This is a band of deep convection and associated precipitation oriented in
northeast/southwest direction over large parts of tropical and subtropical Brazil and the Atlantic Ocean. In its active phase





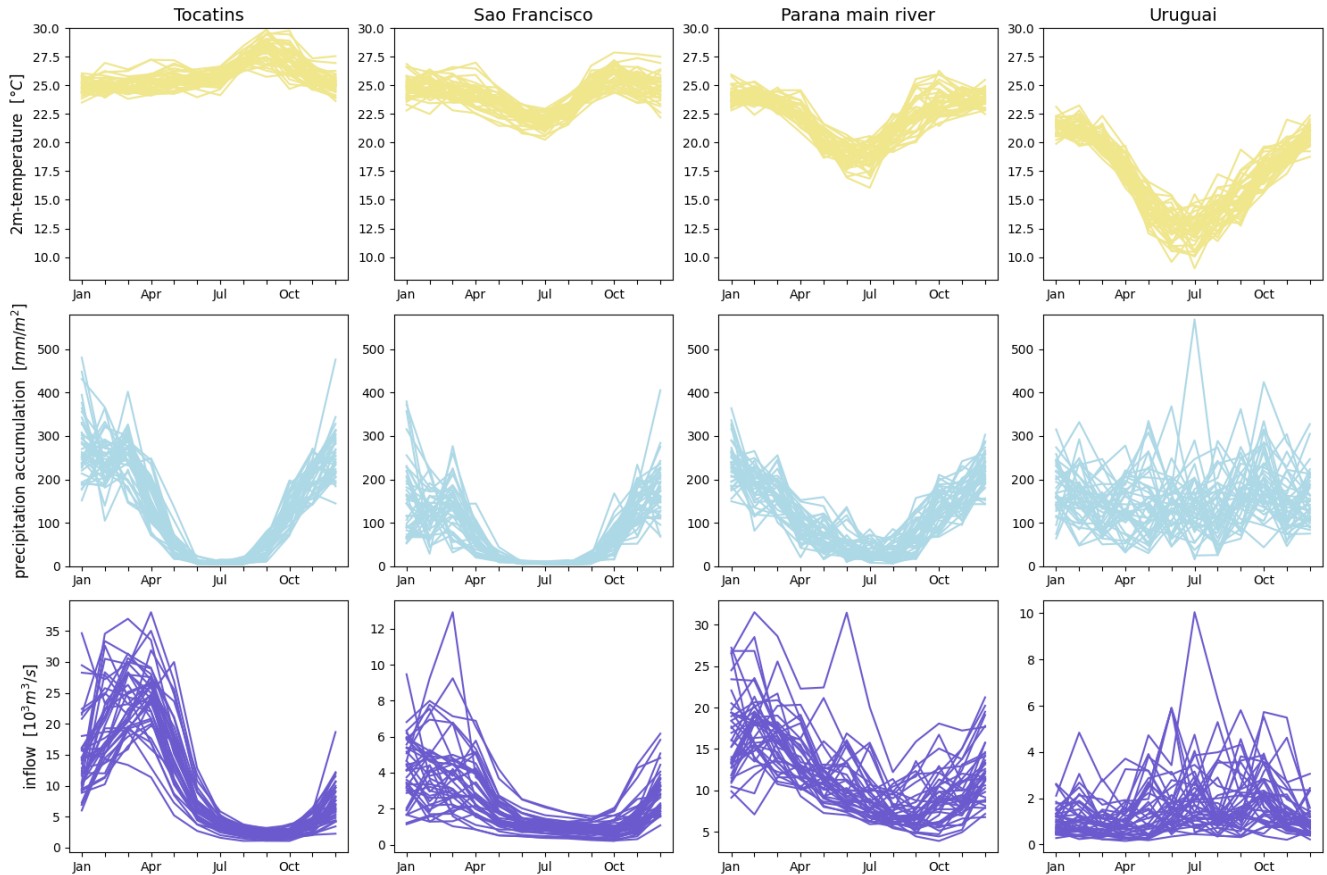

**Figure 2.** Monthly ERA5 temperature averages, CHIRPS precipitation accumulations and streamflow for four selected catchments. Each individual curve corresponds to one year during the 1981-2020 period.

during austral summer, especially between December and February, it brings large amounts of rainfall to Central Brazil (Rosa et al., 2020). In southern Brazil, rainfall originates from synoptic systems, and both rainfall and streamflow is distributed more evenly over the year.

Figure 2 depicts temperature, precipitation and streamflow series from catchments in different parts of Brazil, and gives an idea of the respective annual cycles. The Xingu catchment located in the central-northern part of Brazil receives substantially less rain between May and September, and with a 1-month lag this is also the low water season for this catchment. The annual cycles look similar for the Tocatins catchment in the north-eastern inland of Brazil, with very little rain and corresponding reduced streamflow during austral winter. A much less pronounced but otherwise similar annual cycle is seen for the Parana main river catchment in the central-south, while both precipitation and streamflow in the Uruguai catchment in southern Brazil vary more across different years than across different seasons. In contrast to the catchments further north, however, we see a pronounced seasonal cycle of average temperatures.





**Figure 3.** Correlation coefficients of monthly streamflow anomalies and CHIRPS precipitation anomalies at different time lags.

## 2.5 Lagged correlations between precipitation and streamflow

Among the meteorological variables available as output from the CMIP6 models, precipitation amounts and temperature were considered the most important ones. Especially for the larger catchments the concentration time, i.e., the time it takes for precipitation that falls in the catchment to arrive at the outlet, can be on the same order or longer than the monthly aggregation time scale considered here. Moreover, without a hydrological model that keeps track of antecedant soil moisture conditions, precipitation anomalies in preceding months may be an important factor determining streamflow. Figure 3 depicts the correlation coefficients of monthly streamflow anomalies and monthly precipitation anomalies at different time lags and at different times of the year. The plots confirm that precipitation anomalies during the preceding month can be equally important predictors in particular catchments and seasons, and that conditions further back can also have an impact. This will be considered in the construction of predictors used in the statistical model described below.





## 3   Methods

In this section we describe the construction of a statistical model used to link monthly average temperature and accumulated precipitation over each catchment to the associated streamflow. The exploratory analysis shown above suggested that precipitation at different time lags is an important predictor in any such model. In addition, temperature is included due to its

close connection with evapotranspiration, i.e., the sum of evaporation and transpiration by plants, which both reduce runoff. Specifically, the following predictors for monthly streamflow are considered:

1. Concurrent precipitation amounts

2. Precipitation amounts during the preceding month

3. Total precipitation accumulation 2-4 months prior to the month of interest

4. Concurrent monthly average temperature

These choices are based on the insights gained from Figure 3 and try to balance model flexibility with the need to avoid an overly complex model with too many parameters. It is clear though from this figure that the model must be able to adapt to the season and each particular catchment. The following subsections describe the technical details of how this can be accomplished.

### 3.1   Data standardization

As a preliminary step, both predictand (streamflow) and the predictors specified above are standardized. If we denote by $y_{m,c,i}$ the streamflow observation from month $m$, catchment $c$ and year $i$, the corresponding standardized streamflow anomaly is given by:

$$\tilde{y}_{m,c,i} = \frac{y_{m,c,i} - \hat{\mu}_{m,c}}{\hat{\sigma}_{m,c}}, \tag{1}$$

where $\hat{\mu}_{m,c}$ is the mean monthly streamflow for catchment $c$, and $\hat{\sigma}_{m,c}$ is the corresponding standard deviation. The predictors are standardized in the same way, and concurrent and lagged/aggregated precipitation anomalies are denoted by $\tilde{P}_{m,c,i}$,

$\tilde{P}_{m-1,c,i}$, and $\tilde{P}_{m-2/3/4,c,i}$, respectively, while concurrent monthly average temperatures anomalies are denoted by $\tilde{T}_{m,c,i}$.

Working with standardized anomalies has three major benefits:

1. It acts as an implicit bias correction when the regression model is applied to climate model simulations,

2. It permits a meaningful comparison of regression parameters across months and catchments since systematic spatial and seasonal differences in the amplitude of the original variables are removed, and

3. It allows one to omit the intercept parameter from the regression model.

To see the first point, consider a typical bias correction strategy for climate model simulations (e.g., Ho et al., 2012) in the basic form where the distributions of the model and observation climatology have the same shape but possibly different means and





standard deviations. For a given month, year, catchment, and weather variable, say 2-m temperature, we omit the corresponding subscripts $m, c$ and $i$ from the notation, and denote by $\mu_{mod}, \sigma_{mod}, \mu_{obs}, \sigma_{obs}$ the climatological means and standard deviations

of the model and observations, respectively. The bias-corrected value $T_{bc}$ of a temperature value $T_{mod}$ simulated by a climate model is then obtained via

$$T_{bc} = \mu_{obs} + \frac{\sigma_{obs}}{\sigma_{mod}}(T_{mod} - \mu_{mod}).$$

By rewriting this to

$$\frac{T_{bc} - \mu_{obs}}{\sigma_{obs}} = \frac{T_{mod} - \mu_{mod}}{\sigma_{mod}},$$

we see that the standardized anomaly of $T_{bc}$ relative to the observation climatology is identical to the standardized anomaly $T_{mod}$ relative to the model climatology. If a statistical model based on standardized anomalies $\tilde{P}_{m,c,i}, \tilde{P}_{m-1,c,i}, \tilde{P}_{m-2/3/4,c,i},$

and $\tilde{T}_{m,c,i}$ calculated from ERA5 and CHIRPS data is applied to climate model simulations that are standardized with respect to their own climatology, the above equations show that this is equivalent to working with bias corrected (against ERA5 and CHIRPS data) climate model output. This is a big advantage in the light of results reported by Eden et al. (2014), who suggest that climate model simulations from general circulation models (GCMs) are competitive with those from regional climate models (RCMs) in a setup where both are bias corrected. Standardization as described above thus opens the door to employing

the more widely available GCM simulations without clear detriments regarding the quality of the resulting projections.

### 3.2 Constrained linear regression

Additional scatter plots (not shown here) of the four predictors listed above against the associated streamflow values do not suggest that their relation is extremely complex or non-linear, so given the objective of a fully interpretable model, multiple linear regression is a natural choice. With the standardized data from Section 3.1, this model takes the form:

$$\tilde{y}_{m,c,\cdot} = \beta_{m,c,1} \cdot \tilde{P}_{m,c,\cdot} + \beta_{m,c,2} \cdot \tilde{P}_{m-1,c,\cdot} + \beta_{m,c,3} \cdot \tilde{P}_{m-2/3/4,c,\cdot} + \beta_{m,c,4} \cdot \tilde{T}_{m,i,\cdot} + \varepsilon_{m,c,\cdot}, \qquad (2)$$

with regression coefficients $\beta_{m,c,1}, \beta_{m,c,2}, \beta_{m,c,3}, \beta_{m,c,4}$ specific to each catchment and month, and residuals $\varepsilon_{m,c,\cdot}$ representing the year-to-year variability of streamflow anomalies not explained by the predictor anomalies.

The simple form in (2) permits a clear understanding of how the streamflow anomalies depend on the different predictors: a positive regression coefficient implies that a positive predictor anomaly translates into a positive streamflow anomaly, while a negative regression coefficient translates a positive predictor anomaly into a negative streamflow anomaly. This allows one

to constrain the regression coefficients based on our physical understanding. For all three precipitation-based predictors, positive anomalies should entail an increase in streamflow, while negative anomalies should entail a decrease in streamflow. For temperature, on the contrary, we expect positive anomalies to go along with enhanced evapotranspiration and thus reduced streamflow. These constraints can be imposed on the model by requiring:

$$\beta_{m,c,1} \geq 0, \quad \beta_{m,c,2}, \geq 0 \quad \beta_{m,c,3}, \geq 0, \quad \text{and} \quad \beta_{m,c,4} \leq 0.$$





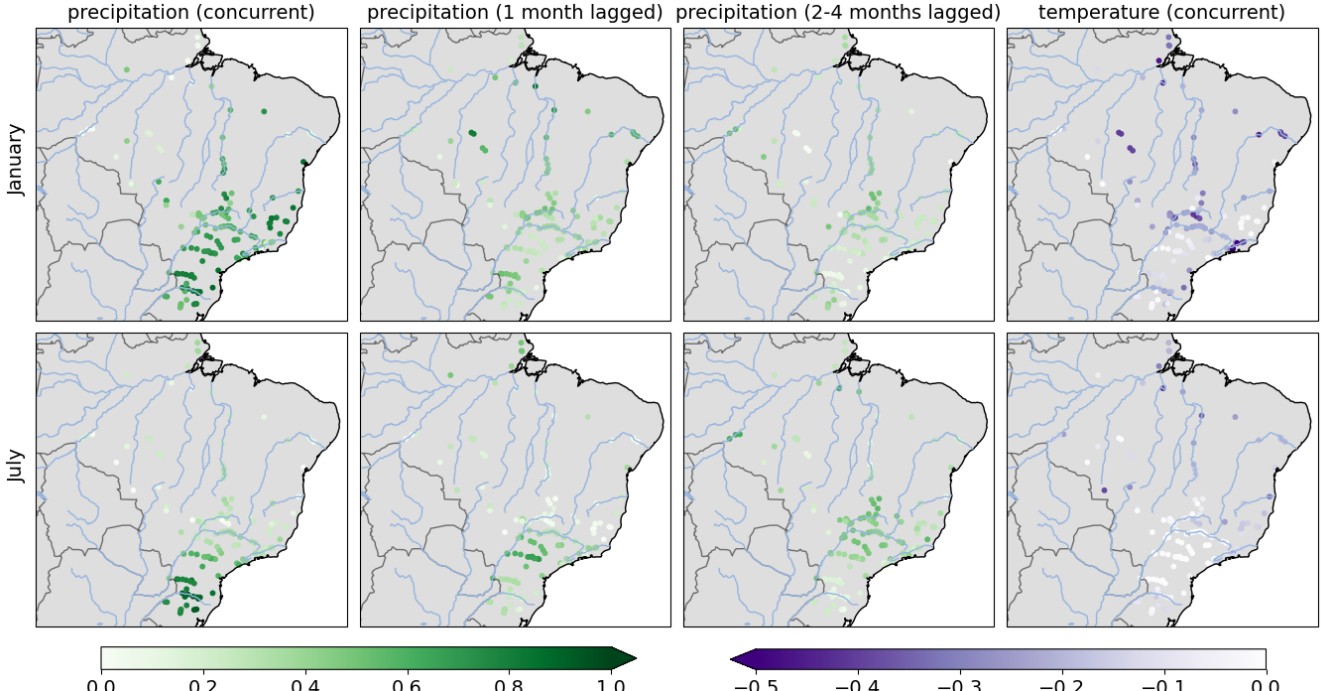

**Figure 4.** Regression coefficients for January and July, estimated via constrained least squares estimation separately for each catchment and month.

They provide some minimal regularization of the regression model and prevent physically implausible predictor-predictand
relationships that might otherwise arise due to collinearity of the different predictors and overfitting. Such effects were seen
in preliminary experiments where unconstrained linear regression was tested, sometimes resulting in streamflow projections
that increased dramatically with increasing temperature as a function of time. The above coefficients can be estimated by
minimizing the least squared residuals $\varepsilon_{m,c,\cdot}$, where minimization, due to the constraints, must be performed by an optimizer
like CVXOPT (Andersen et al., 2011).

Figure 4 depicts the regression coefficients estimated with the procedure described above. Due to the standardization, their
magnitude also has a direct interpretation and reflects the relative importance of the associated predictor. In accordance with
Figures 2 and 3, this importance varies both spatially and seasonally. In July, for example, concurrent precipitation anomalies
are by far the most important predictor in southern Brazil, while catchments in central Brazil rely more on the precipitation
anomalies a few months earlier to explain inter-annual streamflow variability.

Some patterns seen in Figure 4, however, are somewhat questionable from a hydrological perspective. In central and eastern
Brazil, temperature coefficients for January differ substantially even over short distances and with no apparent connection
to catchment size. Consider, for example, the subcatchents of the Corumbá and Araguari river in central Brazil for which
we have highlighted the corresponding gauge locations in Figure 1 in black. These subcatchments are in close proximity





while their temperature coefficients for January are 0.0 and -0.41, respectively. This would imply no sensitivity to temperature

changes at all for the Corumbá subcatchment, while in the Araguari subcatchment a 1° increase of temperature relative to the climatological mean (with precipitation kept fixed at the climatological mean) would entail a 21.2% reduction of inflow. We feel that it is physically implausible that the impact of evapotranspiration on streamflow would be so spatially sporadic, and we find the magnitude of implied streamflow changes concerning given the intended use of this model to project future streamflow based on climate model output.

A likely cause of these physically unrealistic patterns is overfitting of the respective regression models. If decreasing streamflow trends in some subcatchments within the 1981-2020 period, for example, are not sufficiently explained through the other predictors, the regression model may erroneously attribute them to a general warming trend as expressed through large negative temperature coefficients for these catchments. For the precipitation predictors, the spatial patterns in Figure 4 are more plausible, though upon closer inspection one can also find examples of small scale variability that may caused by overfitting rather

than differences in climatology. In the subsequent subsection we discuss a variant of the regression model that aims to retain its flexibility to adapt to regional and seasonal differences in climatology while suppressing some of the spurious variability of the regression coefficients seen in Figure 4.

### 3.3 Modeling seasonal and regional patterns through neural network embeddings

In order to prevent overfitting the coefficients of the regression model (2), some suitable way of sharing information across

seasons and regions has to be found while still allowing the coefficients to vary across these dimensions. Traditionally, spatial statistical models like the INLA framework (Rue et al., 2009) are used for such a task, but those require certain structural assumptions on the type of spatio-temporal covariability and can become rather complex for a multi-variate regression problem like the one studied here. The advent of user-friendly machine learning libraries like PyTorch (Paszke et al., 2019) has opened up the alternative avenue of using neural networks for this purpose, and this approach will be explored in the following.

#### 3.3.1 General idea of the model

The type of neural network used here, a multilayer perceptron (MLP), consists of a sequence of *layers* that each perform a linear transformation of its input followed by a nonlinear activation function. If each input is connected with each output, the layer is called fully connected or *dense*. Through the repeated application of nonlinear activation functions the MLP is capable of representing rather complex functional relationships between its inputs ('features') and the prediction target ('labels', here:

streamflow anomalies). A disadvantage of the multilayer structure is that the learned functional relationships are rather non-transparent and permit little understanding of how the model arrived at its conclusion. Here, we avoid this by using a somewhat unconventional neural network architecture in which the actual predictors (temperature and lagged precipitation anomalies) never pass through any nonlinear function and therefore retain a linear relation with the prediction target. In contrast to the constrained regression framework discussed in Subsection 3.2, however, the regression coefficients for each catchment and

each month are estimated simultaneously and obtained as a complex, nonlinear function of a (arbitrary but unique) catchment ID and month ID. This is achieved through so-called *embeddings*, mappings from a categorical variable to a vector of real





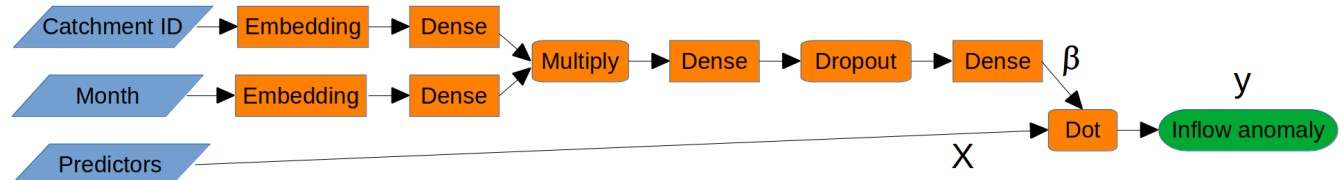

**Figure 5.** Schematic of the neural network proposed as an alternative approach to estimating the regression coefficients for each catchment and month.

numbers encoding information about that variable in an abstract form. Abstract, because this representation is not necessarily connected to any physical space, and inferred purely from the input and output data, though in our case we may for example expect that the embedding of the catchments is connected to their geographical location and possibly to their size.

### 3.3.2 Neural network architecture

Figure 5 illustrates the proposed neural network architecture in a schematic. The output data is the same as in (2), while the input data now consists of catchment ID and month in addition to the four meteorological predictors. These additional, categorical inputs are embedded into separate real vector spaces from where they each pass dense layers whose output is then multiplied pointwise. One may think of the combination of catchment embedding and dense layer as a component that learns a number of relevant spatial patterns which are then weighted and combined based on information about the month associated with the respective input. The resulting vector passes through two further dense layers and a so-called *dropout* layer, which randomly masks components of the input vector during the neural network training process and thereby helps prevent overfitting (Srivastava et al., 2014). The last dense layer produces a four-dimensional output that will be interpreted as the vector $\boldsymbol{\beta}_{m,c} = (\beta_{m,c,1}, \beta_{m,c,2}, \beta_{m,c,3}, \beta_{m,c,4})'$ of regression coefficients to be multiplied to the four meteorological predictors in the same way as in (2). Here, no explicit constraints are imposed on the four coefficients since we find the information sharing across catchments and seasons to be suffcient to prevent physically implausible predictor-predictand relationships like those seen in Figure 4.

### 3.3.3 Model training

Both dense and embedding layers depend on a (relatively large) number of model parameters ('weights') that determine the particular data transformation performed in these layers. These are inferred from the data in a training process in which we minimize a mean squared error loss function, similar to the constrained regression framework in Subsection 3.2, except that we now use the Adam optimizer (Kingma and Ba, 2014) commonly used in connection with neural networks. For more details about the training process for neural networks see e.g. Goodfellow et al. (2016).





### 3.3.4 Hyperparameters

In addition to the model parameters, several *hyperparameters* have to be determined that define the specific neural network architecture and the training process. These include choices like the particular activation function used within the dense layer, the batch size, i.e., the number of samples considered in each iteration of the neural network training process, and the learning rate, i.e., the step size that the optimizer makes during each iteration while seeking to reduce the training loss. We determined those three hyperparameters by monitoring the training progress made with different choices in some test cases, and ended up

choosing exponential linear unit (ELU) activation functions (Clevert et al., 2015), a batch size equivalent to one year of training data, and a learning rate parameter of $0.005$. There are several other hyperparameters defining the components of the neural network shown in Figure 5 for which it is not so easy to find good values through some basic exploration:

- the dimension of the embedding space for the catchments

- the dimension of the embedding space for the months

- the number of *nodes* (i.e., the output dimension) in the first dense layer

- the number of *nodes* in the second dense layer

- the *dropout rate*, i.e. the probability with which a connection is masked during training

We determined these parameters through a systematic hyperparameter tuning process described in Appendix A.

### 3.3.5 Early stopping

The primary motivation for embedding regression model (2) into a neural network framework is to prevent overfitting, and in addition to enabling information sharing across seasons and catchments, this framework comes with a variety of measures to accomplish that. One common strategy is to further split the training data set into a training and validation sample and use the latter to evaluate how well the model trained on a different part of the data generalizes to unseen samples. As the training process progresses, the average loss over the training sample decreases, and for as long as the model truly gets better the

average loss over the validation sample decreases as well. A validation loss that stops decreasing or even increases is a sign of overfitting, and when this is detected the neural network training is terminated. This strategy is referred to as *early stopping*, and was used here to save computation time and ensure that the fitted model generalizes well across different combinations of catchments, months, and across the 1981-2020 training period. The training-validation split was performed by diving the data set into four folds where the first fold contains the years 1981, 1985, ..., 2017, and the other folds are shifted each by

one year. One fold is then used for validation, the remaining three are used for training. This entails four different, possible training-validation splits, and we fit a separate neural network to each of them, calculate the resulting regression coefficients $\beta_{m,c}$ for each catchment and month, and use the mean over the four sets of regression coefficients as an alternative to the coefficients obtained through catchment- and month-wise constrained least squares estimation discussed in Subsection 3.2.





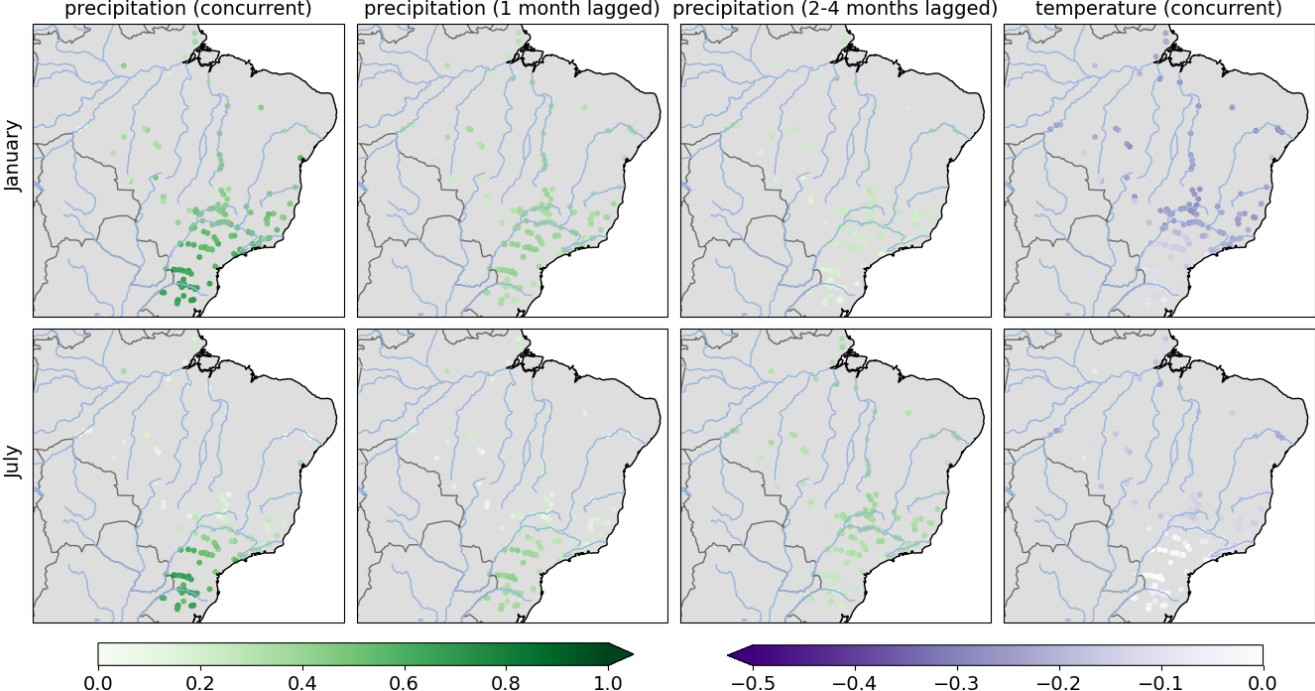

**Figure 6.** Regression coefficients for January and July, estimated with the neural network approach using embeddings to model their dependence on the catchment and month.

### 3.3.6 Model interpretation

The particular architecture of the neural network model proposed here makes it that the output of the last dense layer in the schematic in Figure 5 can be interpreted as a vector of the same regression coefficients in (2) that were previously fitted within a constrained regression framework. We can therefore look at these coefficients (see Figure 6) and compare them directly to those depicted in Figure 4. While the general spatial and seasonal patterns seen in these two figures are similar, the neural network based regression coefficients are not subject to the spurious small-scale variations seen in Figure 4. Their spatial

smoothness is quite remarkable in so far as the neural network did not receive any explicit information about the location of each catchment, and no prior assumption about homogeneity within different subregions has been made. Even though we have not imposed explicit constraints on the coefficients, all precipitation coefficients are positive (i.e., increased precipitation entails increased streamflow) and all temperature coefficients are negative (i.e., higher temperatures entail more evaporation and decreasing streamflow) in line with our physical intuition. The estimated temperature coefficients for the Corumbá and

Araguari subcatchment in January (see discussion in Section 3.2) are now -0.25 and -0.28, respectively. These values imply a decrease in inflow by -14.5% and -14.3%, respectively, if temperature increases by 1° relative to the climatological mean (with precipitation kept fixed at the climatological mean).





The spatially more plausible patterns of the regression coefficients come at the expense of their magnitude though, which is somewhat dampened compared to Figure 4, and might imply that less inter-annual streamflow variability is explained through the meteorological predictors. Whether this is indeed the case will be examined in the next section.

## 4 Results

The ultimate purpose of the statistical models proposed in Section 3 is to apply them to climate model output in order to obtain multi-decadal streamflow projections. This requires that a sufficiently large fraction of inter-annual streamflow variability can be explained through meteorological predictors simulated by climate models. We check this before generating and discussing the resulting streamflow projections.

### 4.1 Coefficients of determination

To evaluate how well inter-annual streamflow variability is explained not just within the data set to which the model is fitted but also for hitherto unseen years, a slightly different protocol for parameter estimation is adopted. For the results presented in this subsection, a leave-one-year-out cross validation approach is applied to the 40 years of available data, i.e. one year $i$ is held out at a time, the respective models are fitted/trained with data from the remaining 39 years, used to predict streamflows during the left-out year, and the prediction error $\varepsilon_{m,c,i}$ is recorded for each catchment and month. This procedure is repeated for all 40 years, and the cross-validated *coefficients of determination* are calculated as

$$R^2_{cv,m,c} = 1 - \frac{\sum_{i=1}^{40} \varepsilon^2_{m,c,i}}{\sum_{i=1}^{40}(y_{m,c,i} - \hat{\mu}_{m,c})^2}.$$

The early stopping and hyperparameter optimization for the neural network have to be adapted to the leave-one-year-out cross validation protocol, too. This is done via a training-validation split of the remaining 39 years at a ratio of 2:1 with every third year being used for validation, and a separate hyperparameter optimization (described in Appendix A) for each of the 40 left-out years. The results is a fully out-of-sample evaluation of the respective models' ability to explain streamflow through meteorological predictors, visualized in Figure 7 for one month from each season.

We note that the patterns for both statistical models are extremely similar, despite noticeable difference in the regression coefficients depicted in Figures 4 and 6, and draw two main conclusions:

1. The ability to explain interannual streamflow variability or lack thereof is more due to regional characteristics than due to the particular statistical model. For example, both models struggle in Central Brazil during austral winter, when precipitation amounts are minimal and streamflow is driven by other factors not included in these models.

2. The dampening of the regression coefficients in Figure 6 relative to those in Figure 4 does not entail overall lower coefficients of determination. The larger (in magnitude) regression coefficients obtained with the constrained regression approach may entail more explained variability *in-sample*, but this does not transfer to unseen years.





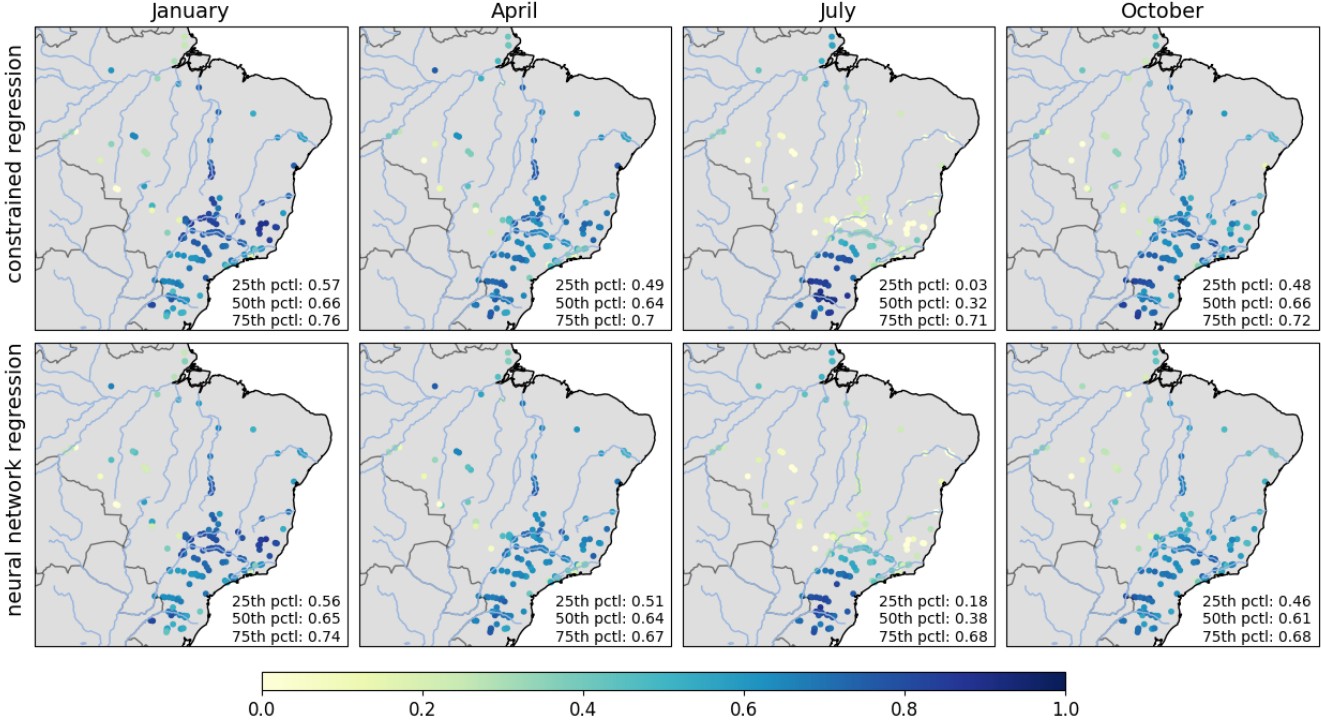

**Figure 7.** Fraction of inter-annual streamflow variability explained (out-of-sample) by the constrained regression model and the neural network regression approach. The inset numbers represent the 25th, 50th, and 75th percentile of the values across all catchments for a given month.

The small differences one can observe are in favor of the neural network regression approach, typically in catchments/seasons with low $R^2_{cv,m,c}$ like the Rio Grande in July, where the constrained regression model is more prone to overfitting due to the low signal-to-noise ratio, and the information sharing across catchments and months achieved by the neural network approach is most beneficial. Yielding physically more plausible patterns of regression coefficients and comparable or even improved $R^2_{cv,m,c}$, this is the approach we choose to employ for making multi-decadal streamflow projections.

## 4.2 Streamflow projections

To obtain projections of future streamflow, the climate model simulations are processed in the same way as the CHIRPS and ERA5 data in Section 3.1, i.e., the same four meteorological predictors are calculated and standardized similar to (1), with mean and standard deviation calculated over the same 1981-2020 period and separately for each catchment, month, and climate model. Systematic biases of climate model output (seen, e.g., in Firpo et al., 2022, Figure 6) are removed through the standardization of this output with respect to each model's own climatology, as explained in Section 3.1.

Figures 8 and 9 depict the resulting projections for different months and subcatchments of the Uruguai and Paranaiba catchment, respectively. Streamflow data were available back to 1960, so we also show the historical CMIP6 simulations




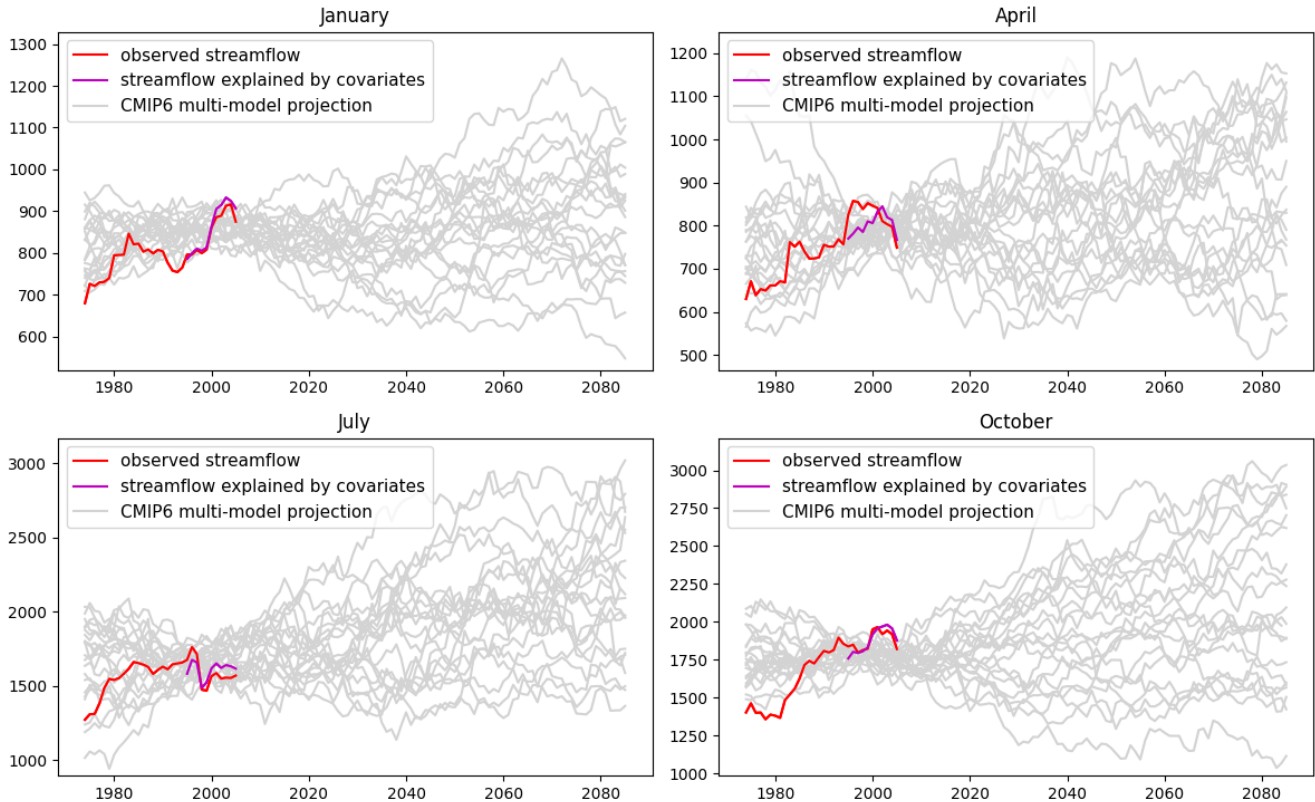

**Figure 8.** Historical and simulated 30-year moving average streamflows for a subcatchment of the Uruguai catchment in southern Brazil for different months. The 'fitted values' are the 30-year moving average streamflow predictions by the statistical model when applied to the CHIRPS- and ERA5-based predictors it was trained with.

back to that year. To filter out some of the year-to-year variability, centered 30-year moving averages of all curves are shown.

The different scenarios - one for each climate model - give an idea of the range of possible outcomes. We note though that this is not a probabilistic forecast in any strict sense as several other sources of uncertainty are not accounted for in these plots (see discussion in Section 5). One of these sources of uncertainty is the unexplained part of the interannual streamflow variability, which is quite large for example in July in the subcatchment of the Paranaiba shown in Figure 9. In this plot, the large unexplained interannual streamflow variability manifests in a poor agreement of the observed streamflow with the values

predicted by the CHIRPS and ERA5 based covariates. With the regression models used here, a low $R^2_{cv,m,c}$ tends to go along with projections that are too conservative, i.e. they underestimate trends in streamflow and do not sufficiently represent the internal variability of streamflow on decadal time scales, thus making it more likely for the historical observed streamflow curve to be outside the range of historical simulated streamflows. In most of the other plots in Figures 8 and 9, the fitted curves match the observed streamflow much better and thus indicate a lesser degree of statistical model uncertainty. Whenever



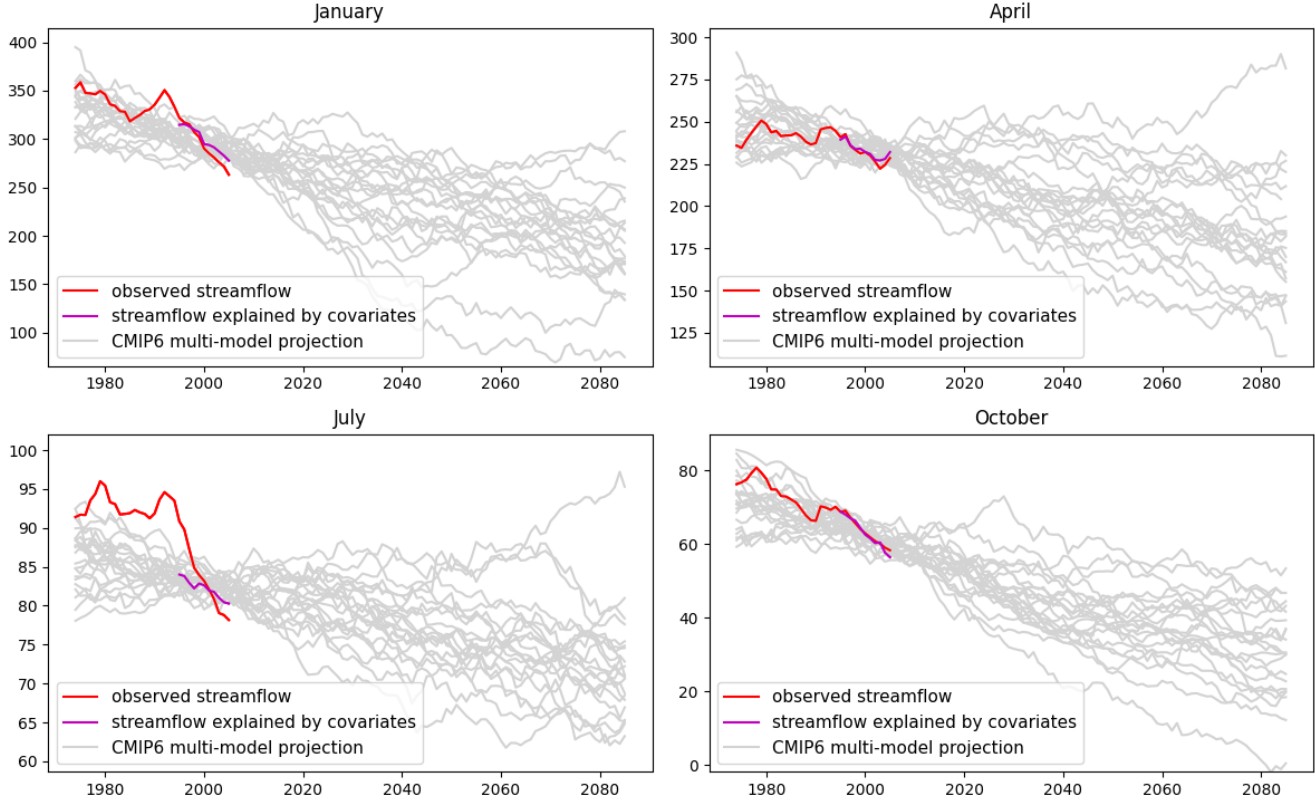

**Figure 9.** Same as Figure 8 but for a subcatchment of the Paranaiba catchment in the midwest/southeast part of Brazil.

this goes along with a clear trend in the CMIP6 multi-model output, this trend translates into a trend of anticipated future streamflow, seen e.g. in the October panel of Figure 9. The trends in the CMIP6 multi-model simulations over southern Brazil are less pronounced, and we therefore only see relatively weak trends in Figure 8, despite generally good model fits. This discussion illustrates that Figure 7 provides important context for the interpretation of the projections discussed here and helps determine how much confidence we should have in the streamflow projections for each catchment and month.

To get an overview over projected changes in streamflow across all catchments, we calculate, for each climate model simulation, the relative change of simulated streamflows between a reference period 1991-2020 and two future periods, 2021-2050 and 2036-2065. The median change across the 22 CMIP6 models for different months is depicted in Figure 10. From the discussion above we recall that in regions and seasons where the $R^2_{cv,m,c}$ of the statistical model is low, the magnitude of change tends to be underestimated. Yet, some clear patterns emerge that are in line with projected hydroclimatological changes

in South America reported e.g. by Marengo et al. (2012) or Zaninelli et al. (2019). Over northern, north-eastern, central, and south-eastern Brazil, a trend towards reduced streamflow is expected for virtually all seasons, especially though for the austral spring and summer season. The lack of a clear change signal during austral winter is at least in part due to a lower $R^2_{cv,m,c}$





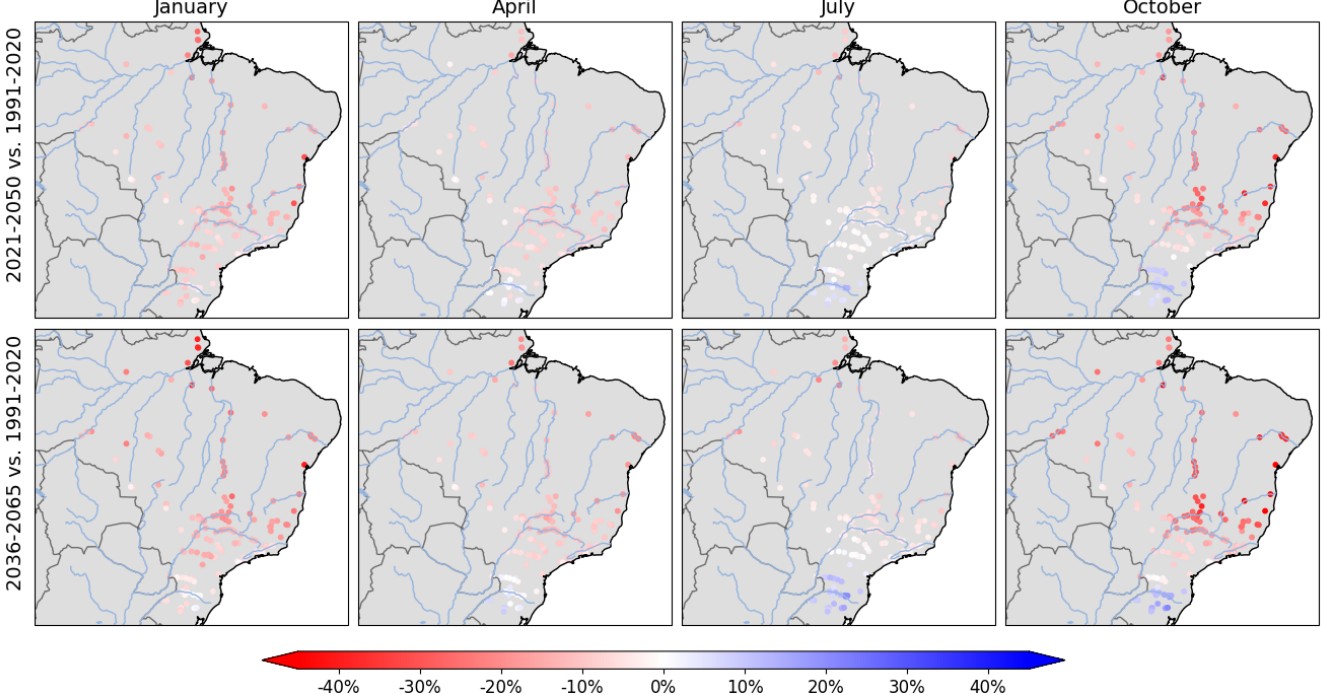

**Figure 10.** Projected change (%) in streamflow between the reference period 1991-2020 and two 30-year periods centered around 2035 and 2050, respectively, for selected months across different seasons.

of the statistical model in that season. Since part of the streamflow during that season originates from rainfall in preceding months (see discussion in Section 2.4), especially in central and north-eastern Brazil, we surmise it is in fact also subject to a decreasing trend that carries over from the preceding seasons. The only region with a projected increase in streamflow is southern Brazil, where e.g. the Uruguai catchment is projected to see a 10.7% (13.3%) increase in streamflow in July and a 10.1% (12.2%) increase in streamflow in October between the 1991-2020 and 2021-2050 (2036-2065) period.

To what degree are different the predictors in our linear model driving these trends? To answer this, we repeat the above calculation for the reference period and the 2036-2065 period with all but one of the four regression coefficients set to zero. The resulting median change signal is then only based on a single predictor, and we can compare the sign and magnitude of that change, depicted in Figure 11, with the changes seen in Figure 10 above. Despite the larger (in magnitude) regression co-efficients of the precipitation based predictors, most of the projected decrease in streamflow is driven by the projected increase in temperature over the next decades and the associated increase in evapotranspiration. Some minor contributions from pro-jected decreases in precipitation can be observed over central and eastern Brazil during the austral spring and summer season. The projected increase in streamflow is southern Brazil, on the contrary, is driven by the projected increase in precipitation over this area. We note that, in addition to the caveat regarding unexplained interannual streamflow variability, this analysis is limited by the simplifying assumption of a linear model structure which does not account for possible non-linear responses of







**Figure 11.** Projected change (%) in streamflow between 1991-2020 and 2036-2065 with all but one predictor in our regression model removed.





the hydrological system to a future climate or interaction between precipitation and evapotranspiration. Given the great care though with which we have attempted to prevent overrepresenting the role of temperature in our statistical model, we still find it notable at least qualitatively that this variable is identified as the primary driver of projected changes in streamflow.

## 5 Discussion: Uncertainty of the Projected Changes

The projected changes depicted in Figure 10 represent the median across a range of different climate models. While they are in line with projected hydroclimatological changes in South America reported e.g. by Marengo et al. (2012) or Zaninelli et al. (2019), we want to stress that these numbers are subject to substantial uncertainty arising from several sources:

1. Uncertainty about future atmospheric greenhouse gas concentrations.

2. Uncertainty due to limitations of climate models.

3. Uncertainty due to natural variability in the climate system.

4. Streamflow variability not explained by our statistical model.

Looking at the full range of projections (see examples in Figures 8 and 9) associated with the different climate models gives some idea of the magnitude of internal variability and disagreement between climate models, but the scenarios should not be viewed as an exact probabilistic representation of these sources of uncertainty. The uncertainty about future atmospheric greenhouse gas concentrations cannot be quantified in any objective way. Only the unexplained interannual streamflow variability could be quantified objectively as the residual variance in our statistical model and superimposed on the different climate model projections. Since this would still only capture part of the overall uncertainty, we have chosen not to calculate confidence intervals on that basis and rather encourage readers to consider the $R^2_{cv,m,c}$ values depicted in Figure 7 when drawing conclusions from Section 4.2, as they provide important context for the uncertainty of the projections related to shortcomings of the statistical model in explaining streamflow variability.

## 6 Conclusions

This paper proposes a linear statistical model that links monthly precipitation and temperature anomalies to anomalies of streamflow and can thus be used in combination with climate model output to obtain streamflow projections in cases where a hydrological model is not readily available. The model overcomes the challenge of a small training sample size by using a neural network framework which estimates the regression parameters for all catchments and all months of the year simultaneously, while retaining the interpretable linear model structure that can easily be checked for physical plausible relationships between temperature, precipitation and streamflow. The model is particularly well-suited for situations where interpretability is a priority, and/or when only short records of monthly streamflow data are available and LSTMs may not perform as effectively.

To demonstrate the proposed model over Brazil, it is applied to the output of 22 CMIP6 climate models to generate multi-decadal streamflow projections over 157 Brazilian catchments. Under the caveat of substantial internal variability that is also





reflected by a large spread between projections by the different CMIP6 models, several trends emerge. Streamflow in northern and central Brazil, where ITCZ and SACZ, respectively, are the main drivers of rainfall, is projected to decrease during all
months in which streamflow is primarily driven by concurrent rainfall. For southern Brazil, on the contrary, streamflow is projected to increase during the austral winter and spring season, while no clear trend is expected for the remaining two seasons. These results are in line with projections of hydroclimatological changes in South America reported previously.

The framework proposed here allows one to translate projections of meteorological conditions into projections of stream-flow. Those can be used, for example, for projections of hydroelectric power production and thereby help inform allocation
of resources. Its conceptual simplicity entails that additional, possibly non-stationary factors like land use, deforestation, or possible feedbacks in drying trends through increased water use are not considered. This can reduce the model's ability to explain a major fraction of interannual streamflow variability, especially during seasons with limited rainfall. However, the simple form makes it easy to transfer the methodology to others regions on the globe and apply it to any set of catchments for which streamflow data is available.

*Code and data availability.* All data used in this study are publicly available through the following websites:

CHIRPS-2.0: https://data.chc.ucsb.edu/products/CHIRPS-2.0

ERA5: https://cds.climate.copernicus.eu/datasets

Streamflow: https://www.ons.org.br/topo/acesso-restrito

CMIP6: https://aims2.llnl.gov
For details of exactly which data sets have been downloaded and how they were pre-processed see Section 2. Python code to reproduce the different steps of the analysis presented here is provided at https://github.com/SeasonalForecastingEngine/BrazilStreamflow/

## Appendix A: Hyperparameter tuning

We use the open-source, automated hyperparameter optimization framework Optuna (Akiba et al., 2019) to efficiently explore the search space of candidate hyperparameters (see Table A1) which determine the specific architecture of the neural network
model proposed in Section 3.3. The optimization was performed in the leave-one-year-out cross-validation setup of Section 4.1, i.e. a separate set of optimal hyperparameters was determined for each left-out year 1981-2020 with a 2:1 split of the remaining years into training and validation data. In addition to permitting a rigorous assessment of the coefficients of determination of the resulting regression models, this approach yields an entire distribution of hyperparameters and thereby insights into the sensitivity of the model performance to the particular choice of hyperparameters. Given the large overlap of data used for the
different cross-validation folds, a highly dispersed distribution indicates that the specific hyperparameter value is not all that crucial. A tight distribution, on the contrary, indicates that certain values are particularly conducive to good model performance.

Figure A1 shows histograms of the selected values across the 40 years. It suggests that only for the catchment embedding dimension there is a very clear preference for a particular value, namely 6, the largest value within the tested range. For the dropout rate, an intermediate value of 0.3 tends to give the best results but there is significant spread around that value. Similarly,





**Table A1.** Candidate values for the hyperparameters to be optimized, and value selected for the model used to generate the streamflow projections

| hyperparameter | candidate values | selected value |
|---|---|---|
| embedding dimension for catchments | 3, 4, 5, 6 | 6 |
| embedding dimension for months | 1, 2, 3 | 2 |
| number of nodes in the 1st hidden layer(s) | 10, 15, ..., 40 | 25 |
| number of nodes in the 2nd hidden layer | 10, 15, ..., 40 | 20 |
| dropout rate | 0.0, 0.1, ..., 0.5 | 0.3 |

for the month embedding dimension, smaller values tend to perform better, but not by a huge margin. For the number of nodes in the hidden layers, there is no clear tendency at all. As a result of this analysis and the conclusion that model performance is not overly sensitive to the particular choice of hyperparameters, we use the optimized values only within the cross-validated setting of Section 4.1. For the neural network used in Section 4.2 to generate streamflow projections we just use fixed values, shown in the last column of Table A1, instead of running a new Optuna hyperparameter optimization for the four different

training-validation splits of that setting.





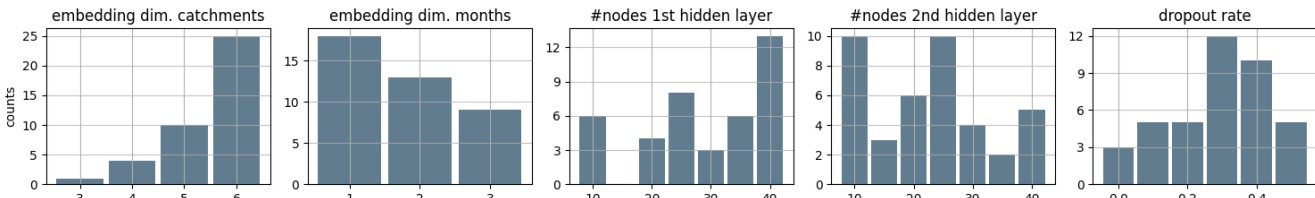

**Figure A1.** Histograms of the optimal hyperparameters selected by Optuna for each of the 40 cross-validated years.

*Author contributions.* **Michael Scheuerer**: Methodology, Formal analysis, Software, Writing - Original Draft. **Emilie Byermoen**: Data Curation, Writing - Review & Editing. **Julia Ribeiro de Oliveira**: Data Curation, Writing - Review & Editing. **Thea Roksvåg**: Data Curation, Methodology, Formal analysis, Writing - Review & Editing. **Dagrun Vikhamar Schuler**: Conceptualization, Data Curation, Writing - Review & Editing.

*Competing interests.* The authors declare that they have no known competing financial interests or personal relationships that could have appeared to influence the work reported in this paper.

*Acknowledgements.* We acknowledge the World Climate Research Programme, which, through its Working Group on Coupled Modelling, coordinated and promoted CMIP6. We thank the climate modeling groups for producing and making available their model output, the Earth System Grid Federation (ESGF) for archiving the data and providing access, and the multiple funding agencies who support CMIP6 and

ESGF. This work has benefited greatly from insightful discussions with Gilca Palma (Climatempo), Thordis L. Thorarinsdottir (University of Oslo), and Gastón Santisteban-Martinez, Asgeir Petersen-Øverlier, Knut Sand, and Ida Eggen (Statkraft Energi AS). It was supported by the Research Council of Norway through grant 309562 ('Climate Futures').



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
