# Peer review of "Multi-decadal Streamflow Projections for Catchments in Brazil based on CMIP6 Multi-model Simulations and Neural Network Embeddings for Linear Regression Models"

_EGUsphere, 2025_

## Author Response (AR1)

We thank all reviewers for their useful and constructive comments. These comments have been addressed as already announced in the open discussion of our manuscript. Below is a summary of this discussion with the specific comments and our responses. In the revised version of the manuscript, we have added some references including those suggested in a community comment. We have updated Figures 1, 8, and 9 according to the reviewers' suggestions. A marked-up version of the revised manuscript with all changes made to the main text highlighted in red/blue will also be provided.

**Reviewer 1:**

- I suggest revising the study area map, adding coordinates, north arrow, scale.
Thanks for your suggestions, we'll add coordinates (which should also give an idea of the scale) and north arrow in the revised version of the manuscript.

- Some literature reviews can be added about hybrid statistical-physical models in introduction.
We'll add some references about hybrid statistical-physical models in the revised version of the manuscript.

- You are using different data sources CHIRPS vs. ERA5, did you do some sensitivity check analyses?
Yes. We had originally used ERA5 data for both temperature and precipitation, but had then discovered discrepancies between historical trends for ERA5 precipitation and streamflow anomalies. Encouraged by comparisons of ERA5 and CHIRPS precipitation data over Brazil reported in the literature, we tested CHIRPS data as an alternative to ERA5 data and obtained better results with CHIRPS (then still using only the constrained regression framework). The downside of this change, a reduction in the number of years available for fitting the model, led to the development of the neural network regression approach.

- Have you think about physical relationships between temperature and vapor pressure of water (like considering clausius-claperyon equation)?
In earlier stages of the project, we have discussed and tested more sophisticated ways to model the impact of temperature changes (due to global warming) on streamflow through increased evapotranspiration, which also take interactions between temperature and precipitation changes into account. However, these attempts with more complex (non-linear) approaches were not successful, likely due to the limited data and often poor signal-to-noise ratio, and we therefore decided to move forward with a simple linear model.

- Considering nearest grid point, did you use orographic effects?
No, we have not corrected for orographic effects. While discrepancies between model grid and real orography may indeed entail biases for both temperature and precipitation, we use the same rationale that we describe in section 3.1 ('Data standardization') in the context of possible biases in climate model simulations: by using only standardized anomalies in our model, systematic biases are corrected implicitly as they cancel out in the standardized anomalies. Of course, this rationale does not work for non-linear bias effects, but as explained above, such effects are too complex to identify and correct in a robust way with the available data.

- I suggest adding skill scores as well for climatology.
By 'skill scores', are you referring to the fraction of explained variability depicted in Figure 7? This quantity is effectively a mean squared error (MSE) skill score, and climatology in this context would be constant prediction of zero anomaly (since the climatological signal is removed in the standardization described in section 3.1). So, by definition, the explained variability (MSE skill score) for climatology is zero everywhere.

**Reviewer 2:**

- Could the authors clarify why the CatchmendID pipeline was separated from the Month pipeline in the neural network architecture? A brief explanation would be helpful.

We have experimented with alternative architectures where the catchment and month information is combined at an earlier stage. The performance was very similar, the reason why we chose the architecture with separate pipelines in the article is that we like the interpretation briefly explained in section 3.3.2 as one pipeline learning a set of spatial patterns and the other pipeline learning how to weigh these patterns differently over the course of the year. One can visualize this as in the figures attached to this comment, where we depicted the first 4 (of 25) spatial patterns for the first cross validation fold (see section 3.3.5) and the associated month-specific coefficients. We find the possibility to visualize and interpret the result of the embeddings in this way an advantage of our chosen architecture. If interpretability is the main focus, further improvements could be achieved by
a) choosing activation functions after the respective first dense layer that entail non-negative coefficients, and
b) reducing the number of nodes in the first hidden layer (output dimension of these dense layers) which could potentially lead to more unique individual patterns while likely (see figure A1) only having a minor negative impact on the model's performance.

**Reviewer 3:**

- The authors provide an overview of process-based hydrological models and data-driven approaches, including LSTMs. One potential improvement is to elaborate slightly more on why existing hydrological models could not be adapted in this context.

Using hydrological models is certainly possible, and the first community comment has pointed us to a paper that calculates projections of future streamflow over South America using that approach. However, this requires both expertise with the respective local hydroclimate and a substantial amount of time to calibrate that model for all catchments. Since our project partners at Statkraft want to use this model in several different parts of the world and often have to provide a first iteration of future streamflow simulations rather quickly, there was a desire for an approach that can more easily be transferred to different regions. This was one of the primary motivations for this work, and we will expand our explanations to make this more clear.

- In the introduction, the authors emphasize the limitations of LSTM models, characterizing them as "black boxes." However, in this study, LSTMs or neural networks are not applied to make direct predictions but rather to learn coefficient embeddings for a linear regression model. Moreover, there are many established approaches to improve the interpretability of neural networks. I suggest that the authors compare existing explainability methods (e.g., attention mechanisms, feature attribution techniques) with the embedding approach adopted in this study, to more clearly situate the method within the broader context of explainable machine learning.

We will expand this section and add some comments and references on explainability and interpretability. In our understanding, explainability methods can help better understand the sensitivity of the output to the various inputs, but cannot make it interpretable to the same degree as a process-based hydrological or linear statistical model where one has a clear, intuitive understanding of the model's decisions. We'll also add a sentence that makes it clear where our proposed method is situated w.r.t. explainable machine learning approaches.

- Please briefly explain how missing months or gaps in time series were handled in the regressions.
In our setup, missing values only occurred in the context that for a small number of combinations of months and catchments, the streamflow data looked suspicious (identical values across the majority of years). In those cases, we removed the entire month-catchment combination from the analysis. The case where only a few years for a given month-catchment combination are missing did not occur in our study, but should not pose any problems as long as the standardization (section 3.1) can be calculated in a robust way. The regression model (especially the one fitted within a neural network framework using month and catchment embeddings) can be fitted with the missing years removed from the training data set.

- Figure 5 is helpful, but it would improve understanding to provide a clearer description of the dimensionality of embeddings and dense layer outputs in the main text rather than in the appendix.
We will provide that information in the main text in discussing Figure 5 to give an idea about the typical hyperparameter values early on, while still referring to the 'Hyperparameter' subsection and the Appendix for the technical details of how these values were obtained.

- Figure 7 shows the percentiles of the coefficients of determination across all catchments. It is noted that in July, the model performance is relatively poor, with many catchments having values even below 0.03. Please check whether this is correct, and provide an explanation for this phenomenon. How this affects the robustness of long-term projections.

We believe that this is correct, and is a result of typically low precipitation amounts in July over large parts of Brazil (see Figure 2), which makes modeling the rainfall-runoff relation more difficult than in other months, and more dependent on long-term storage mechanisms and possibly other factors (more impact from reservoir operations that was not fully accounted for in the available streamflow series). We will expand the discussion of the negative implications for long-term projections and associated uncertainty (section 5), and

add a sentence to the conclusion to make it clear that this is where more complex approaches may have the most potential for improvement.

- The horizontal and vertical axis labels and units in Figure 8 and Figure 9 are missing. The horizontal axis label (year) should be self-explanatory, for the vertical axis we will add labels and units to the two left panels in the revised version.

- Figure 11 analyzed the predictor contributions which is insightful. It is noted that this decomposition does not capture interaction effects, which could be a limitation. For example, the temperature emerges as the primary driver of projected declines in streamflow. However, this may be partly an artifact of the linear model structure. Have the authors tested alternative formulations, such as including interaction terms (e.g., temperature × precipitation) or exploring nonlinear relationships?

In earlier stages of the project, we have discussed and tested more sophisticated ways to model the impact of temperature changes on streamflow through increased evapotranspiration which also take interactions between temperature and precipitation changes into account. However, these attempts with more complex (non-linear) approaches were not successful, likely due to the limited data and often poor signal-to-noise ratio, and we therefore decided to move forward with a simple linear model. We agree though that there is a potential danger of an omitted variable bias with our model, and we will add this caveat to the discussion of Figure 11.

- While smoothing year-to-year variability is understandable, applying centered 30-year moving averages can mask decadal shifts and dampen trends, particularly in non-stationary time series. It could be informative to provide supplementary figures with alternative window lengths (e.g., 10 or 20 years) or without smoothing, to demonstrate the stability of trends.

We will provide supplementary figures with the suggested alternative window lengths.